# Dry Powder Comprised of Isoniazid-Loaded Nanoparticles of Hyaluronic Acid in Conjugation with Mannose-Anchored Chitosan for Macrophage-Targeted Pulmonary Administration in Tuberculosis

**DOI:** 10.3390/pharmaceutics14081543

**Published:** 2022-07-25

**Authors:** Mahwash Mukhtar, Noemi Csaba, Sandra Robla, Rubén Varela-Calviño, Attila Nagy, Katalin Burian, Dávid Kókai, Rita Ambrus

**Affiliations:** 1Institute of Pharmaceutical Technology and Regulatory Affairs, Faculty of Pharmacy, University of Szeged, 6720 Szeged, Hungary; mahwash.mukhtar@szte.hu; 2Department of Pharmacology, Pharmacy and Pharmaceutical Technology, University of Santiago de Compostela, 15782 Santiago de Compostela, Spain; noemi.csaba@usc.es (N.C.); sandra.robla@outlook.es (S.R.); 3Center for Research in Molecular Medicine and Chronic Diseases, University of Santiago de Compostela, 15782 Santiago de Compostela, Spain; 4Department of Biochemistry & Molecular Biology, School of Pharmacy, University of Santiago de Compostela, 15782 Santiago de Compostela, Spain; ruben.varela@usc.es; 5Wigner Research Centre for Physics, 1121 Budapest, Hungary; nagy.attila@wigner.hu; 6Department of Medical Microbiology, Albert Szent-Györgyi Medical School, University of Szeged, 6720 Szeged, Hungary; burian.katalin@med.u-szeged.hu (K.B.); kokai.david@med.u-szeged.hu (D.K.)

**Keywords:** dry powder inhaler, immune regulation, inhalation, isoniazid, mannose conjugation, macrophage phenotype, next-generation impactor, tuberculosis

## Abstract

Marketed dosage forms fail to deliver anti-tubercular drugs directly to the lungs in pulmonary Tuberculosis (TB). Therefore, nanomediated isoniazid (INH)-loaded dry powder for inhalation (Nano-DPI) was developed for macrophage-targeted delivery in TB. Mannosylated chitosan (MC) and hyaluronic acid (HA) with an affinity for the surface mannose and CD44 receptors of macrophages were used in conjugation to prepare hybrid nanosuspension by ionic gelation method using cross-linker, sodium tri-polyphosphate (TPP) followed by freeze-drying to obtain a dry powder composed of nanoparticles (INH-MC/HA NPs). Nanoformulations were evaluated for aerodynamic characteristics, cytotoxicity, hemocompatibility, macrophage phenotype analysis, and immune regulation. Cellular uptake imaging was also conducted to evaluate the uptake of NPs. The nanopowders did not pose any significant toxicity to the cells, along with good compatibility with red blood cells (RBCs). The pro-inflammatory costimulatory markers were upregulated, demonstrating the activation of T-cell response. Moreover, the NPs did not show any tolerogenic effect on the macrophages. Furthermore, confocal imaging exhibited the translocation of NPs in the cells. Altogether, the findings present that nano-DPI was found to be a promising vehicle for targeting macrophages.

## 1. Introduction

Tuberculosis (TB) remains one of the main causes of death globally, which are estimated to be 1.2 to 1.4 million per year according to WHO, despite the advancement in therapeutics and diagnostics [1]. TB poses a serious socio-economic burden on developing and under-developed countries. Out of all the reported TB pathologies, pulmonary TB contributes to 80% of pathogenesis [2]. The onset of pulmonary TB occurs after the inhalation of *Mycobacterium tuberculosis* (M.Tb). Though M.Tb, microorganisms can be captured by mucous-secreting goblet cells in most instances, they bypass the mucociliary clearance system and are deposited by phagocytosis as a result of interaction between M.Tb surface lipoarabinomannan and surface mannose receptors of alveolar macrophages (AM). Hence, M.Tb finds AM to be its niche for survival and reservoir for replication due to inhibition of phagosome-lysosome fusion [3]. Moreover, the TB microenvironment also facilitates the growth and survival of M.Tb. TB is spreading at an alarming rate due to a lack of patient-adherent therapeutic options and long-term duration of treatment (6 months) with standard therapy. Moreover, the mycobacteria in low proliferative phases and multi-drug resistant strains need more prolonged treatment of over 24 months. The only available option is the vaccine (Bacille Calmette-Guerin, BCG), but it also fails in people sensitized to M.Tb [4]. Among other challenges are the inadequate delivery of effective anti-tubercular agents to the site of infection along with an off-site accumulation of drugs leading to organ toxicity.

Conventional drug delivery systems fail to deliver anti-tubercular drugs to the alveolar region because of indirect delivery via blood. Consequently, innovative approaches need to be fabricated for effective pulmonary drug delivery without off-target accumulation. As the causative agent resides in the host AM, a suitable inhalation system with an excellent aerodynamic profile must be designed to achieve a targeted delivery, which might reduce the dosage frequency. Inhalable nanosystems have been widely and successfully investigated in the past as well with different compositions. Among them, the polymeric nanosystems are advantageous because of their attribute of encapsulating both the hydrophilic and hydrophobic drugs, controlled release profile, desirable pharmacokinetic outcomes, and ability to translocate across the biological barriers [5]. Polymeric nanocarriers comprised of polymers such as alginate, chitosan, [poly (DL-lactide-co-glycolide)] (PLGA), tri-block poly (ethylene glycol) (PEG)-poly (ε-caprolactone) (PCL), etc., have shown promising outcomes in the delivery of nanoparticles (NPs) to the lungs with localized drug release over a long time with minimal cytotoxicity and good therapeutic outcomes [6,7,8,9,10,11]. For this purpose, nanotechnology-based dry powder for inhalation (nano-DPIs) can be a promising opportunity that needs extensive research for bench-to-bedside availability.

Surprisingly, the registered patents (US20200289667A1, US20170319699A1) have already been exploiting the potential of NPs to target macrophages in different diseases. Moreover, the phagocytic feature of AM can be exploited by developing therapies targeted at intra-macrophage infections such as TB. This property can be probed by altering the physical, chemical, and surface characteristic of the NPs [12]. Thus, here we have developed a nano-DPI system using polymers for the optimistic therapy of TB. Antigen-presenting cells (APCs) such as macrophages have overexpression of surface receptors such as CD44 and mannose receptors that can be targeted by developing the nanosystems constituted of polymers serving as a ligand to these receptors. Mannose receptors have a pivotal function in the regulation of adaptive immune response. It is proposed that mannose receptor-mediated endocytosis of mannose conjugated nanoparticles can stimulate an enhanced immune response [13,14].

Based on the idea of targeting the surface receptors of macrophages, suitable polymers were chosen. Hyaluronic acid (HA) is an immune-compatible polymer that also protects against pulmonary injury [15] with an affinity for the CD44 receptors on the macrophages. HA also performs a vital role in the growth of epithelial cells and macrophages [16]. Similarly, chitosan (CS) is derived from marine origin and demonstrates no toxicity to human cells along with biodegradability. Hence, it was the polymer of choice for mannose conjugation to synthesize mannosylated chitosan (MC). Oligosaccharides with terminal mannose on microorganisms can bind to the macrophage mannose receptor and allow their endocytic transport in the cells. This can be correlated with the use of mannose-conjugated polymeric nanoparticles for the intra-macrophage delivery of anti-tubercular drugs by exploiting surface mannose receptors for a T-cell immune response [17]. The use of MC not only facilitates mannose-targeted drug delivery but also promotes controlled drug release. Hence, in this work, we describe the pulmonary drug delivery nanopowder (the powders obtained after drying of the nanosuspensions) to potentially reach the infected AM by using biodegradable polymers. DPIs are propellant-free and cost-effective drug delivery devices for the pulmonary administration of drugs for local or systemic infections. The developed nanopowder was characterized in terms of the aerodynamic parameters and powder morphology. The nanopowder composed of MC and HA was compared with the CS and HA nanopowder in this study. The sole purpose of this comparative study was to evaluate if the synthetic polymer (MC) has any peculiar off-results in terms of cytotoxicity and T-cell pathway stimulation as compared to the natural CS.

## 2. Materials and Methods

### 2.1. Materials

Isoniazid (INH) [IUPAC: Isonicotinylhydrazide] from Pannon Pharma Kft (Hungary). Sodium hyaluronate equivalent to 1.5–1.8 × 10^6^ Da (Hyaluronic acid) from Contipro Biotech (Czech Republic). Chitosan (CS) [75–85% deacetylated, low molecular weight, 50–190 kDa, Poly(D-glucosamine)] and 4′,6-diamidino-2-phenylindole (DAPI) dye from Sigma-Aldrich (St. Louis, MO, USA). Macrophage Raw 264.7 and A549 cell lines were obtained from ATCC (Manassas, VA, USA). Granulocyte-macrophage colony-stimulating factor (GM-CSF), Allophycocyanin (APC)-conjugated anti-human CD83 (CD83-APC), and Phycoerythrin (PE)-conjugated anti-human CD80 (CD80-PE) were purchased from Miltenyi Biotec (Bergisch Gladbach, Germany). Ficoll-Paque ^TM^ PLUS (density 1.077 g/mL) was purchased from GE Healthcare Bioscience AB (Chicago, IL, USA). Dulbecco’s modified Eagle’s medium (DMEM) and Roswell Park Memorial Institute medium (RPMI-1640) were purchased from GIBCO^®^ (Thermo Fischer Scientific, Bedford, MA, USA). Sodium tri-polyphosphate (TPP) from Alfa Aesar (Thermofischer, Munich, Germany). Trifluoroacetic acid (Merk Schuchard OHG), 3-(4,5-dimethylthiazol-2-yl)-5-(3-carboxymethoxyphenyl)-2-(4-sulfophenyl)-2H-tetrazolium (MTS) cell proliferation assay kit was provided by BioVision (Milpitas, CA, USA). Sodium dodecyl sulfate (SDS), 3-(4,5-dimethylthiazol-2-yl)-2,5-diphenyltetrazolium bromide (MTT), and 4-Dimethylamino benzaldehyde were obtained from Sigma-Aldrich (Chemie GmbH, Steinheim, Germany). Fetal Bovine Serum (FBS) and PSG (100 u/mL penicillin, 0.1 mg/mL streptomycin, and 2 mM L-glutamine) were purchased from Invitrogen(Carlsbad, CA, USA). Glacial acetic acid (GAA) was purchased from Molar Chemicals Kft (Hungary). For in vitro experiments, sterile and autoclaved materials were used. All the chemicals were of high purity or reagent grade.

### 2.2. Synthetic Procedure

MC polymer was synthesized by our previously reported method [18]. The synthetic procedure is, however, briefly mentioned in the Appendix A. The number of mannose groups on MC was quantified after synthesis. In short, the polymer was hydrated and put into the 96-well plate. Totals of 20 µL of resorcinol, 100 µL of sulfuric acid, and 50 µL of pristane (2,6,10,14-tetramethyl-pentadecane) were added to each well. The well plate was kept at 90 °C for 20 min, and the optical density (OD) was read by a microplate reader (PerkinElmer, Waltham, MA, USA) at 450 nm.

### 2.3. Characterization of Polymer

The synthesized polymer MC was characterized by Fourier Transform infra-red spectroscope (FTIR) (Thermo Nicolet AVATAR 330, Waltham, MA, USA). IR spectra were acquired by the KBr disc method at 4 cm^−1^ resolution at the wavenumber range of 400–4000 cm^−1^ at room temperature (RT). Moreover, the ^1^H NMR spectroscopy (Bruker BRX-500) was performed in deuterated DMSO to analyze the polymer.

### 2.4. Preparation of Nanoparticles

For the preparation of nanoparticles, the polymer MC was solubilized in 0.5 M glacial acetic acid solution, and HA was dissolved in water. The pH of the polymeric suspension was maintained at 4.9. Following the ionic gelation technique, TPP (0.5–2 mg/mL) was used as a cross-linker to facilitate the ionic interaction between the positively charged amino groups of MC and the anionic charge on HA [19]. TPP was added dropwise to the polymeric suspension consisting of HA and MC. After thorough stirring, INH (10 mg, 10% of oral dose) was added dropwise to the polymeric nanosuspension. The nanosuspension was probe sonicated followed by stirring overnight to obtain a uniform consistency. A similar procedure was used to develop the CS-based nanosuspensions. Rhodamine-B (Rh-B)-labeled NPs were obtained by replacing INH with the fluorescent dye and later dialyzed against deionized water for three days to remove the unattached dye.

### 2.5. Freeze-Drying to Obtain Nanopowders

The prepared nanosuspensions were freeze-dried in Scanvac, Coolsafe 100-9 prototype apparatus (LaboGeneApS, Lynge, Denmark) to obtain the dry powders for inhalation. Four percent trehalose (*v*/*v*) was added to the liquid samples before lyophilization. The pressure of the chamber was maintained at 0.01 mbar throughout the process. Table 1 shows the process parameters recorded over time using a computer program attached to the instrument.

### 2.6. Particle Size, Polydispersity Index (PDI), and Surface Charge

Parameters such as particle size, polydispersity index (PDI), and zeta potential were analyzed by the Malvern zeta sizer Nano ZS (Malvern instrument, Worcestershire, UK). The nanopowders were redispersed in purified water before analysis. All experiments were performed in triplicate and are expressed as mean ± SD.

### 2.7. Encapsulation Efficiency (EE)

The encapsulation efficiency (EE) of the nanosuspensions was evaluated by an indirect method using a supernatant. The supernatants were collected after centrifuging the nanosuspensions at 15,000 g for 30 min. The obtained supernatants were spectrophotometrically analyzed at 264 nm, and Equation (1) was used to calculate the % EE. The percentage of drug loading (DL) was also determined, which is the percentage of the actual mass of the drug loaded in the nanopowder to the total acquired mass of the nanopowder, as given by Equation (2)
% Encapsulation Efficiency = (Total drug-Free drug)/(Total drug) × 100 (1)
% Drug Loading = (Mass of drug loaded in NPs)/(Total mass of NPs) × 100 (2)

### 2.8. Morphological Examination

Nanopowders were studied for their surface morphology by using Scanning Electron Microscopy (SEM) (Hitachi S4700, Hitachi Scientific Ltd., Tokyo, Japan) at 2.0–5.0 kV. Throughout imaging, the pressure of the air was maintained at 1.3–13.0 mPa.

### 2.9. Colloidal Stability at Storage Conditions

The nanosuspensions before freeze-drying were kept at working and storage temperature (25 °C) for 1 month. The average particle size and PDI were determined after certain time points.

### 2.10. In Vitro Aerodynamic Profile by Next-Generation Impactor (NGI)

Next-Generation Impactor (NGI) (Copley Scientific Limited, Nottingham, UK) setup was employed for the assessment of the aerodynamic profile of the freeze-dried sample, INH-MC/HA NPs. Only the mannosylated sample was tested and compared with the results from previous studies by our group. This is the standard method for the determination of the size distribution of particles dispensed from DPIs on collection trays based on aerodynamic size. An optical microscopic method was conjoined with an impactor for the evaluation of particles deposited on each collection tray [20]. Figure 1 shows the measurement setup, which was used for the testing with the NGI. In the measurements, an in-house developed breath simulator generated the breathing waveform (red arrows), an induction port acted as the upper respiratory tract, and a vacuum pump (HCP5 High-capacity pump; Copley Scientific Ltd., Nottingham, UK) with a critical flow controller (TPK 2000; Copley Scientific Ltd., UK) maintained the constant flow along the blue arrows, which delivered the particles from the DPI to the impactor. The compressor compensated for the losses in the system. The mixing inlet (Copley Scientific Ltd., Nottingham, UK) provided the interface between the flow, which activated the DPI, and the main flow that delivered the particles to the NGI device. The NGI determines the aerodynamic size distribution of the particles by the impaction method. The sample flow rate of the NGI was maintained at 90 L/min, which was regularly checked during the measurements with a TSI 4000 thermal mass flow meter [20]. The direction of airflow indicated by arrows is similar to that of the working of the Aerodynamic Particle Sizer (Appendix A). The nanopowder equivalent to 10 mg INH (10% of the recommended oral dose) was loaded into the hydroxypropyl methylcellulose (HPMC) capsules (transparent, size 3, ACG) and was dosed through the Breezhaler^®^ (Novartis) dry powder inhalator device. After the experiment was run, the fine particle fraction (FPF < 3) was calculated, which represents the percentage of particles smaller than 3 µm and denotes the settling of particles in the deeper parts of the lungs. The mass median aerodynamic diameter (MMAD) was also determined, which is defined as the median particle diameter of the particles settled in the NGI. This was evaluated by the logarithmic aerodynamic diameter of the particles between stages 2 and 3 versus the interpolation of the percentage undersize [21]. In general, MMAD is the cut-off diameter at which 50% of the deposited particles are smaller or larger by mass.

Moreover, the aerodynamic size was analyzed by time-of-flight measurements in an accelerated flow through an Aerodynamic Particle Sizer (APS-TSI 3321, Shoreview, USA). As a breath simulator, we used an in-house developed pulmonary waveform generator. It uses a piston pump driven by a programmable logic controller (PLC)-controlled servo motor to generate the inhalation and exhalation air flows (Appendix A). The inhalation volume spans from 0.1 to 6800 cm^3^. The time resolution of the inhalation profile can be set to 20, 50, and 100 ms. The inhalation waveform programmed into the breathing simulator was generated according to the literature (Appendix A) for the measurements [22,23].

### 2.11. Isolation of Monocytes and Differentiation into Macrophages

After informed consent was obtained, the heparinized blood was collected from the healthy donors. The identity of the donors was kept anonymous. The buffy coats were donated by the Organ and Blood Donation Agency (ADOS; Santiago de Compostela, Spain). The Ficoll density gradient separation method was employed to isolate peripheral blood mononuclear cells [24]. In brief, blood was poured into a 50 mL tube in the laminar flow cabinet. The blood was diluted with PBS (1:1) at maintained RT. This diluted blood was carefully added to Ficoll-Paque^TM^ PLUS at a blood/Ficoll ratio of 2:1. Human peripheral blood mononuclear cells (PBMC) were isolated after centrifugation (Allegra X-12R, Beckman Coulter) at 400 g for 30 min at RT on deceleration mode. The upper layer was discarded, leaving behind the PBMC layer, which was carefully transferred to 50 mL centrifugation falcon tubes. The PBMC layer was washed with PBS using centrifugation at 300 g for 10 min to improve the purity by removing the remaining Ficoll media. The obtained cells were then resuspended in an R_2_ medium (RPMI-1640 supplemented with 2% heat-inactivated FBS and 1% of PSG). A total of 10 mL of the cells was seeded into a 75 cm^2^ cell culture flask for 2 h (37 °C, 5% CO_2_) by maintaining the density of cells at 1.2 × 10^6^ cells/mL. After this time, the non-adherent cells, peripheral blood lymphocytes, were washed with PBS and the attached monocytes were cultured for 3 days in R_10_ media (RPMI-1640 supplemented with 10% heat-inactivated FBS and 1% of PSG). After 3 days, the media was replaced with R_10_ media (RPMI-1640 supplemented with 10% heat-inactivated FBS, 1% of PSG, and cytokines (GM-CSF at 100 ng/mL)) for the differentiation of monocytes to macrophages.

### 2.12. Cytotoxicity Studies

Before experiments, the cell lines were cultured in Dulbecco’s Modified Eagle Medium (DMEM) supplemented with 10% (*v*/*v*) fetal bovine serum (FBS) and 1% (*v*/*v*) penicillin-streptomycin-glutamine (PSG). The culture media was replaced after every 2–3 days to maintain the cell confluency. The cultured cells were incubated at 37 °C and 5% CO_2_ in a humidifier chamber.

#### 2.12.1. Cytotoxicity on A549 Cells

MTT assay was performed for the evaluation of the cytotoxic effect of NPs on the cells. A549 cells (adenocarcinoma human alveolar basal epithelial cells) were used as a model for alveolar type II cells, as these cells predominantly constitute the pulmonary alveolar epithelia [25]. For this purpose, A549 cells were seeded and cultured at a density of 4 × 10^4^ cells/well in 96-well culture microplates. Later, the formulations, namely INH, blank CS/HA NPs, blank MC/HA NPs, INH-CS/HA NPs, and INH-MC/HA NPs, were added in different concentrations to the cultures and incubated for 24 h at 37 °C. After incubation for 24 h, 20 μL of MTT was added to each well, and the culture plates were again incubated for 4 h. Next, 100 μL of sodium dodecyl sulfate solution (10% in 0.01 M HCl) was added to the well plates to dissolve the formed formazan crystals. OD was measured using an EZ READ 400 ELISA reader (Biochrom, Cambridge, UK) at 550 nm (ref. 630 nm). Untreated cells with 100% viability were used as a control. All the evaluations were performed in triplicate. The relative cell viability was calculated using the following Equation (3),
Cell viability (% control) = (Absorbance of sample)/(Absorbance of control) × 100(3)

#### 2.12.2. Cytotoxicity on Raw 264.7 Cells

Again, an MTT assay was performed to evaluate the cytotoxic effect of the formulations on the Raw 264.7 cells, which were cultured in the sterile flat-bottom 96-well tissue culture plates at a density of 1 × 10^4^ cells/well for 24 h. The culture media was replaced with the different concentrations of nanopowder samples (100 μL final volume, dissolved in supplemented media) on the following day. Following incubation for 24 h, 10 μL of MTT (dissolved in phosphate buffer saline (PBS)) at 5 mg/mL concentration) was added to the wells. Well culture plates were incubated for 4 h at 37 °C without light. Afterward, the MTT solution was discarded, and the formazan crystals were dissolved by the addition of 100 μL of acid isopropanol (0.04 N HCl in isopropanol). The OD of the plates was read after 10 min at the wavelength of 570 nm (Reference wavelength 630 nm) using a microplate reader (Synergy H1 Hybrid Multi-Mode, BioTek, Winooski, VT, USA) to determine the cell viability using Equation (2). Untreated cells served as a negative control, and sodium dodecyl sulfate (SDS) was used as a positive control [26].

#### 2.12.3. Cytotoxicity on Human Macrophages

An MTS assay was performed to determine the cytotoxic effect of INH, blank CS/HA NPs, blank MC/HA NPs, INH-CS/HA NPs, and INH-MC/HA NPs on the primary macrophage culture. Monocyte-differentiated macrophages were seeded onto the 96-well microplate at a density of 1.10 × 10^5^ cells/mL and incubated for 24 h. Later, the media was replaced by the different concentrations of the samples and incubated for 24 h under standard sterile conditions (37 °C, 5% CO_2_). A total of 10 μL of MTS reagent was then added to the well plates and incubated for 4 h. The absorbance was then measured at 490 nm using a microplate reader (Synergy H1 Hybrid Multi-Mode, BioTek, Winooski, VT, USA). Macrophages in culture media (0% toxicity) served as a negative control, and SDS was used as a positive control (100% toxicity). Equation (2) was used to calculate the viability of cells.

### 2.13. Confocal Imaging for Visualization of Uptake of NPs in A549 and Raw 264.7 Cells

The uptake of NPs was analyzed through confocal laser scanning microscopy. Briefly, A549 cells and Raw 264.7 cells (8 × 10^4^ cells/mL) were seeded onto the individual Lab-Tek^®^ chambered #1.0 Borosilicate cover glass system (0.8 cm^2^/well). After 24 h, the cell culture media was replaced with 300 µL of Rh-B labeled NPs in a concentration of 10 μg/mL and incubated with cells for 2 h. The cells were washed thrice with PBS, and 4% paraformaldehyde was added to fix the cells and allowed to incubate for another 15 min. The cells were again washed with PBS three times followed by the addition of DAPI (300 µM, 1:500 in PBS) nucleus dye and incubated for 50 min. A549 and Raw 264.7 cells without the addition of formulations were used as controls. Following washing with PBS, the mounting media was added to the chamber, and imaging was performed by confocal microscope (Leica SP5, Mannheim, Germany). Rhodamine Ex: 546 nm/Em: 568 nm; DAPI Ex: 359 nm/Em: 457 nm.

### 2.14. Human Macrophage Phenotype Analysis

NPs were incubated with blood-derived macrophages in a 48-well plate at a final concentration of 10 μg/mL for 24 h. Cells were washed with PBS twice (400 g, 6 min at RT) to remove NPs. Later, the cells were resuspended in PBS and stained with an optimal concentration of different antibodies (CD83-APC and CD80-PE) for 25 min at −4 °C in the dark [27]. The cells were washed with PBS again (400 g, 6 min RT) and resuspended in PBS, and kept on ice until measurement. The level of maturation markers was then quantified by flow cytometry in a BD FACSCalibur cytometer. Flowing software (Cell Imaging Core, Turku Centre for Biotechnology) was used to analyze the data. The data have been shown as the ratio between the mean fluorescence intensity (MFI) of the corresponding markers in macrophages incubated with NPs and the MFI of macrophages incubated in culture media.

### 2.15. Tolerogenic Effect of NPs in Macrophages

2,3-Indoleamine dioxygenase (IDO) expression was assessed in the macrophages after their exposure to NPs. The enzyme, IDO, is involved in the catabolism of tryptophan, which is pivotal for the growth of microorganisms and, therefore, directly influences the T-cell tolerance [28]. Moreover, IDO is an immune-suppressive enzyme in macrophages with the function of catabolizing tryptophan into its metabolite, kynurenine, which is responsible for the apoptosis of Th1 cells in vitro. Hence, the IDO assay quantifies the kynurenine in culture media. The study was conducted to evaluate the tolerogenic response of the NPs on macrophages by using the described methods [29]. Briefly, cells were seeded onto a 48-well plate, followed by their incubation with different formulations in a final volume of 0.5 mL. Four hours before the end of the culture period, 1.25 μL of L-tryptophan (100 μM) was added to the medium. A total of 30% trifluoroacetic acid (2:1 *v*/*v*) was mixed with culture media (obtained after the centrifugation of cells at 10,000 g, 5 min at RT) to precipitate the cell debris in another round of centrifugation with the aforementioned parameters. Ehrlich Reagent was added to acquire supernatant, and absorbance was read using a microplate reader at 490 nm.

### 2.16. Hemolysis Assay

Fresh blood from four human donors was collected in the acid citrate dextrose (ACD)-containing tubes. The blood was washed thrice with PBS by centrifugation at 250 g for 5 min, and the red blood cells (RBCs) pellet was collected while the supernatant plasma was discarded. The obtained RBCs pellet was diluted with PBS, and the RBCs suspension was seeded onto a 96-well plate and incubated with NPs for 4 h and 24 h at 37 °C. Triton-X 100 (1% *v*/*v*) and PBS were kept as positive and negative controls, respectively. The absorbance of the samples was measured at 570 nm using a microplate reader (Synergy H1 Hybrid Multi-Mode, BioTek, Winooski, VT, USA), and % hemolysis was calculated using Equation (4),
% hemolysis = (Absorbance of sample-Absorbance of negative control)/(Absorbance of positive control-Absorbance of negative control) × 100(4)

### 2.17. Statistical Analysis

All the experiments were performed in triplicate unless otherwise stated. All the results are expressed as mean ± standard deviation. GraphPad Prism v.6.01 software (GraphPad Software Inc., San Diego, CA, USA) was used for data analysis. A two-way ANOVA test in combination with Dunnett’s multiple comparisons tests was used to present the difference between donor groups.

## 3. Results

### 3.1. Characterization of Polymer

The mannose groups on the MC polymer were quantified to be 232 ± 13 μM per gram. The NMR analysis (Figure 2) demonstrated the mannose conjugation to CS with a peak at 4.03 ppm (the methylene protons of the mannose sugar) [30]. The signals at 2.404 ppm corresponded to the protons of the CH_2_-group, indicating the linking bridge (acetamido group) between mannose and chitosan by Schiff’s base reductive amination [31]. The peak for the methyl group of the non-deacetylated part of CS was observed at 1.631 ppm, and the amine group of CS corresponded to 0.859 ppm [32].

Moreover, FTIR spectra (Appendix A) presented the IR peak at 3351.69 cm^−1^, corresponding to amide bond stretch in MC as a result of conjugation of mannose to the unmodified polymer. The peak at 1780 cm^−1^ presented COO symmetric stretching, and COO asymmetric stretching was seen at 1200 cm^−1^. Further, the NH_2_ band was observed at 1032.73 cm^−1^, followed by the amide bond formation fingerprint peak in MC at 1100 cm^−1^ [33]. Moreover, the peak of mannose stretch can also be seen at 850 cm^−1^. The peak at 1600 cm^−1^ was characteristic of CO-NH_2_ in CS, whereas the peak at 3500 cm^−1^ demonstrated OH bond widening [34].

### 3.2. Freeze-Dried Nanopowders

The average particle size of the drug-loaded MC/HA NPs was found to be 303 ± 16.2 nm. In the past, nanoparticles within the size ranges of 200–350 nm have shown promising uptake into the macrophages [35,36,37]. Moreover, it has been reported that particles below 250 nm size present reduced uptake by the alveolar macrophages and pulmonary endothelial cells [38].

Besides the particle size, the PDI value, 0.179 ± 0.04, was also found to be promising for the INH-MC/HA NPs in comparison to the CS-based samples. The PDI value of less than 0.2 is considered to be ideal in the case of polymeric drug delivery nanovehicles [39]. Hence, the nanopowder was monodispersed with a narrow size distribution. The positive zeta potential was also considered favorable for the high stability of the nanopowders. Further, the cationic-charged moieties have high intracellular uptake efficiency in the macrophages, followed by pulmonary inhalation [40]. Table 2 enlists some parameters of the nanopowders.

### 3.3. Morphological Examination

SEM micrographs (Figure 3) displayed the smooth morphology of the freeze-dried nanopowders with a narrow size distribution from a working distance of 12.8 and 14.2 mm. Likewise, the NPs were scattered uniformly. Blank-CS/HA NPs presented small patches of aggregation that might have been due to remnants of free TPP [41].

### 3.4. Colloidal Stability

The colloidal stability was evaluated for 1 month to demonstrate the minimum stability for operational purposes (Figure 4). Nanosuspensions demonstrated aggregation in the case of a long storage time. However, the average particle sizes were not significantly altered, which might be the reason for inconsistency in the individual readings. Moreover, the nanosuspensions were not sonicated before evaluations to get the real-time behavior. The PDI of the nanosuspensions was, therefore, increased proportionally to each time interval. However, most of the samples had PDI index values ≤ 0.5, which are considered appropriate for mono-disperse nanosystems [42]. Furthermore, freeze-drying was employed to guarantee long-term stability.

### 3.5. In Vitro Aerodynamic Profile

NGI was used to assess the aerodynamic size distribution of the particles from DPI via Breezhaler^®^. The amount of powder in each stage was determined by the optical method. The mannosylated dry powder sample demonstrated favorable results in terms of mass size distribution. First, the data were obtained from the APS by maintaining setting channel bounds according to the cut-off sizes of NGI plates. Later, the powder was evaluated for mass size distribution by NGI, and results were acquired based on the surface coverage of the collection plates. As shown in Appendix A, the highest fraction of particles in the dry powder system was within the range of 1.37–2.3 µm, as determined by APS with settings according to NGI. Later, the results obtained from NGI (Figure 5) confirm the data from APS, i.e., the size distribution was correlated to the previous measurement, and the average mass size distribution of the particles was within the same range of 1.37–2.3 µm, exhibiting deposition in the peripheral airways (terminal bronchioles and alveoli). The average of all the results (performed four times) is shown in Appendix A. Thirty-five percent of FPF was found to be less than <3, highlighting that this ratio of the nanopowder was deposited in the deeper lung. MMAD was calculated to be 2.7 µm, which explains that a higher proportion of the particles demonstrated good aerodynamic behavior in terms of the surface properties of the particle.

### 3.6. Cytotoxicity Studies

An MTT assay was performed to investigate the cytotoxicity of the nanopowders and nascent INH on A549 and Raw 264.7 macrophages. After an exposure of 24 h with different concentrations of INH and nanopowders (0.01, 0.5, 1 mg/mL), it was evident that the A549 cell viability was pronounced for all samples (Figure 6a). The drug-loaded nanopowders demonstrated more than 80% viability for A549 cells. Blank MC/HA nanopowder showed remarkable results and was 100% in A549 cells. The cell viability for Raw 264.7 macrophages was concentration-dependent (Figure 6b). INH presented more than 50% viability at low concentrations of 0.5 and 0.01 mg/mL, which led to the evident conclusion that at higher doses, the drug is toxic to macrophages. Moreover, the cationic NPs display a high affinity toward macrophages, and hence the toxicity can increase depending on the concentration of NPs [43]. The MTT assay displayed a reduction in Raw 264.7 macrophage viability with increasing concentrations of polymers. However, all the concentrations were found to have more than 50% cell viability.

Likewise, the MTS assay, which was performed to access the impact of nanopowders on the metabolic activity of primary macrophages, revealed similar results. The % cell viability was evidently but not primarily dependent on the increase in the concentrations of the samples (Figure 6c). All the samples presented cell viability of ≥70%. These cytotoxicity data were obtained after 24 h incubation. The main purpose behind the viability studies on the primary cells was to evaluate the accurate concentration of the samples optimal for the human macrophage phenotype analysis.

### 3.7. Visualization of NPs in the A549 and Raw 264.7 Cells

Confocal laser scanning microscopy (CLSM) was employed for the qualitative assessment of the fluorescent-labeled Rh-B NPs. Figure 7 shows the shift in the intensity of fluorescence on the internalization of NPs as compared to the control (untreated cells). Human alveolar epithelial cells, A549, were also used to study the uptake behavior of NPs. It can be seen clearly that the internalization of NPs in the A549 cells was lower compared to the Raw 264.7 macrophages. This might be due to the well-established reason that A549 cells are not responsive to the NPs in a similar way to immune cells (macrophages) [44]. Immune cells, such as macrophages, identify antigens and NPs by phagocytosis, surface receptor-based endocytosis, and micropinocytosis. Supposedly, macrophages most likely responded through the surface receptors and hence translocated the moieties with mannose composition efficiently. The internalization intensity of the NPs was found to be higher in the case of MC/HA NPs as compared to CS/HA NPs (Appendix A), demonstrating the advantage of mannose conjugation to the polymer. Quantification of Rh-B-labeled NPs was also performed by flow cytometry (Appendix A).

### 3.8. Human Macrophage Phenotype Analysis

The expression of T-lymphocyte costimulatory molecules (CD83 and CD80), the indicators of pro-inflammatory-activated phenotypes in macrophages, was evaluated by the incubation of NPs with macrophages for 2 h. The expression was analyzed by flow cytometry (Appendix A). CD83 is elevated and observed in the activated macrophages. CD80 is the prime costimulatory marker affecting cytokine secretion [45]. The delivery of antigen to macrophages upregulates the expression of CD83 and CD80, which are considered to induce T cell receptor signaling and activation. The expression was many folds higher for the MC/HA NPs when compared to other nanoformulations (average MFI = 1 for control). The results in Figure 8a show the relative comparison between the expression of CD83 and CD80 in macrophages derived from three different blood donors. On the whole, the findings demonstrate that blank and drug-loaded MC/HA NPs significantly upregulated the costimulatory markers in comparison to other NPs.

### 3.9. Tolerogenic Activity

IDO expression by macrophages influences peripheral tolerance and immune regulation. IDO assay was performed to determine if the NPs were inducing a tolerogenic phenotype on macrophages characteristic to the suppression of T-cells and the promotion of tolerance (contrary to pro-inflammatory response) (Figure 8b). It was analyzed by quantifying the IDO activity following incubation of NPs with macrophages. The NP samples demonstrated a similar response to that of control macrophages, establishing no tolerogenic effect of the NPs.

### 3.10. Hemolytic Activity

Hemolysis assay demonstrates the biocompatibility of NPs with RBCs to get insight into the behavior of formulations for in vivo applications. The % viability of RBCs was evaluated against the Triton-X (positive control) with 100% cell lysis. All the samples posed no toxicity on the RBCs, indicating biocompatibility with RBCs (Figure 8c). In this case, all the NPs samples were hemocompatible.

## 4. Discussion

The major obstacle in the treatment of TB is the inadequate availability of the drug in the affected organ. Therefore, a dosage form capable of delivering the effective drug concentration to the alveolar region of the lungs is the primary priority for the treatment of TB. In recent times, the use of nanotechnology has gained interest for organ-targeted drug delivery. The use of a nanotechnology-based aerosolization approach can limit off-site drug accumulation. Moreover, the nanocarriers show high residence time in the lungs because of the presence of mucous. Therefore, based on this rationale, dry powder for inhalation was developed in this study using nanotechnology. A hybrid nano-approach was utilized for the fabrication of nano-DPI by using two polymers, MC and HA. These polymers are non-toxic, non-thrombogenic, biodegradable, biocompatible, and non-immunogenic [46]. The limitations, such as mucociliary clearance of a large proportion of inhaled powders and exhalation of small-sized particles, can be overcome by using a ligand anchored polymeric drug delivery system. The ligand anchorage to the NPs can reduce the reticuloendothelial system (RES) uptake and improve the availability of the drug at the target site. Further, as mentioned previously, the mannose receptor is a C-type lectin that can identify the mannose-containing polymers with high affinity. Therefore, the mannosylated polymer was used in this study to improve the drug delivery in TB. Likewise, HA also presents a high affinity for the CD44 receptors on the surface of macrophages [47]. Therefore, these polymers were chosen for developing the macrophage-targeted nanoparticulate system. The cost-effective ionic gelation method was employed to prepare the polymeric nanosuspension, and anti-tubercular INH was loaded into the NPs, followed by freeze-drying with 4% trehalose as a cryoprotectant to yield dry powder. Freeze-drying removes the solvent from samples by sublimation of frozen content in the primary drying step and unfrozen solvents in the secondary drying step. The freeze-drying time is, however, dependent on the product height and hence varies with the sample volume [48].

The average particle size for the INH-MC/HA-loaded nano-DPI was found to be 303 ± 16.2 nm, with a monodisperse nature indicated by a PDI of 0.179. The surface zeta potential was 34.3 ± 6.03, showing good stability of the formulation. The % EE of the INH-CS/HA NPs and INH-MC/HA NPs was high, which might have been achieved because of the synthetic approach used for the NPs. The drug was loaded after the synthesis of NPs and hence was strongly adhered to the voids of the NPs. It has been observed that the use of the ionic gelation method yields NPs with a plexus, and the drug can be embedded within the matrices.

The prediction of the pharmacokinetic and pharmacodynamic (PK/PD) profile for the inhaled drugs can be a complicated protocol because of the complex pulmonary geometry. Hence, testing of the aerodynamic particle size distribution and deposition of the particles by NGI can narrow the gap between the in vitro and in vivo performance testing to accelerate research and development (R&D). The particles in the NGI are driven by the constant airflow towards different stages with defined cut-off diameters. The average mass size distribution was evaluated by NGI, operated according to European Pharmacopeial 2014 requirements. The results demonstrated that a high fraction of the particles had a size range of 1.37–2.3 µm, correlating to the deposition in stages 6 and 7 with the geometrical standard deviation (GSD) of 1.50. GSD determines the variation in diameters of particles within the aerosol cloud. Usually, the GSD values > 1.2 present the heterodispersive nature of the aerosols with broad particle size distribution [49]. The aerodynamic profile can further be improved by using the alternate drying procedure for the nanosuspensions to acquire the powder for inhalation, which has also been demonstrated by us in the past [50].

The cytotoxicity studies revealed that all the samples had high % cell viability and posed no toxicity to A549 cells, Raw 264.7 macrophage, and primary cultures. Likewise, the demonstration of hemolytic activity is essential because of the safety concerns of NPs. The in vitro hemolytic activity on RBCs is evaluated by spectrophotometric analysis of plasma-free hemoglobin derivatives after the incubation of NPs with blood. Centrifugation was performed to remove the undamaged RBCs. As established, the safe hemolytic ratio for biomaterials should be less than 5% according to ISO/TR 7406 [51]. In this study, all the formulations posed no toxicity to RBCs.

Principally, adaptive immune response mediated by T-cells is essential for the control of M.Tb. NPs did not interfere with the adaptive immune response and facilitated the T-cell signaling and activation. By the upregulation of the costimulatory molecule CD80, there is an elevation of interleukin-6 that exhibits pro-inflammatory activity and plays a role in the resistance against TB [52]. Correspondingly, CD83 plays a role in resolving immune responses in TB and is also essential during the differentiation of T-lymphocytes along with maintaining tolerance. The inhibition of CD83 alleviates the inflammation [53]. However, the developed NPs improved the expression of costimulatory markers.

The uptake of nanopowders by the A549 cells and Raw 264.7 macrophages was established by confocal microscopy. The nuclei of the cells were stained with DAPI dye to facilitate visual imaging. A549 cells (primarily comprised of alveolar basal epithelial cells) were used for studying the uptake of the NPs because the inhaled particles interact with the alveolar epithelia before engulfment by macrophages. MC/HA nanopowders presented high localization into the cells as compared to the CS/HA nanopowders. Altogether, the NPs were able to be translocated into the macrophages, which corroborated with the aim of the study. Further, the tolerogenic response was assessed for the nanopowders. The tolerogenic response is responsible for the immunosuppression that contradicts the T-cell response required in TB. Increased IDO activity by the macrophages suppresses effector T-cells and elevates regulatory T-cells, which then promotes immune tolerance [54], which is not favorable for the treatment of TB. The nanopowders did not exhibit any tolerogenic effect, as demonstrated by the IDO assay. All the results were compared with the CS/HA-based nanopowder for a thorough understanding of the various parameters that might otherwise be compromised by using MC polymer. Everything considered, the mannose-anchored nanoparticulate system is suitable for delivering anti-tubercular to the macrophages in the TB.

## 5. Conclusions

In this study, macrophage-targeted nanoparticles were developed to achieve higher retention at the site of a bacterial niche for promising therapy of TB. For this purpose, polymers were chosen because of their affinity for the surface receptor of macrophages for the uptake of encapsulated anti-tubercular inside the immune cells. The nanosystem INH-MC/HA was fabricated to be administered by inhalation via dry powder inhalers for efficient delivery to the lungs. The results reported that the nano dry powders had higher deposition in the deeper region of the lungs followed by pulmonary administration. The ability of NPs to interact with macrophages was conserved and amplified by using mannose-anchored chitosan along with HA. Altogether, nano-DPIs presented promising fundamental outcomes that might impact investigative studies in animals in the future.

## Figures and Tables

**Figure 1 pharmaceutics-14-01543-f001:**
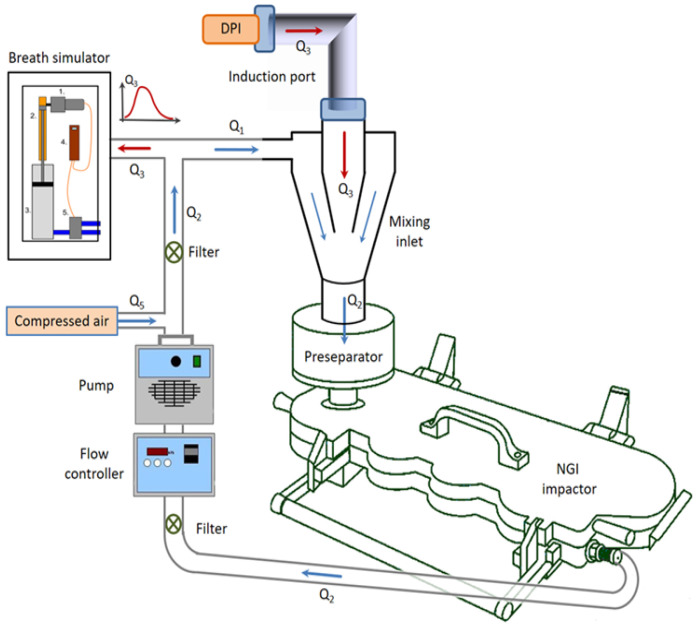
The schematic design and components of the measurement setup: DPI, induction port, NGI, vacuum pump with a critical flow controller, and mixing inlet.

**Figure 2 pharmaceutics-14-01543-f002:**
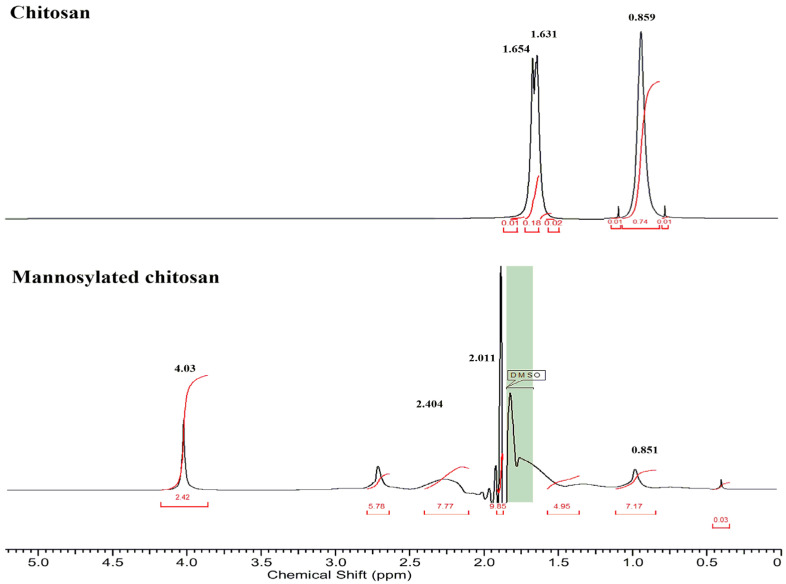
NMR spectra of chitosan and mannosylated chitosan. The respective chemical shift peak signals are mentioned in bold. The red text presents the integral (or area) of the peak signals.

**Figure 3 pharmaceutics-14-01543-f003:**
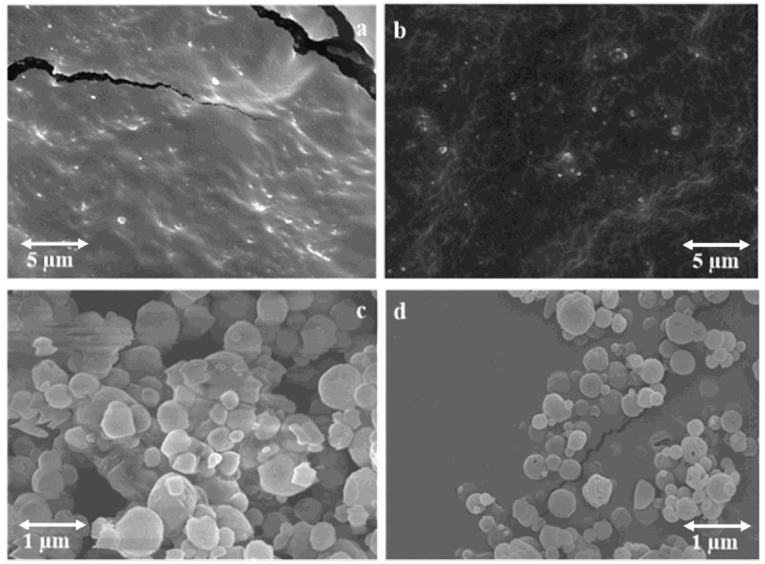
SEM micrographs of INH-CS/HA nanopowder (**a**,**c**) and INH-MC/HA NPs (**b**,**d**) were obtained separately with the 1 and 5 µm scale bars.

**Figure 4 pharmaceutics-14-01543-f004:**
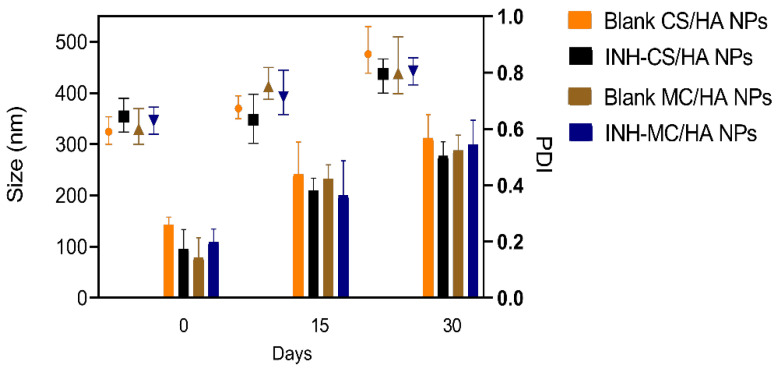
Stability of the nanosuspensions with the relative change in particle size (nm) and polydispersity index (PDI) at 15-day-intervals over one month. Results are expressed as mean ± S.D, performed in triplicate.

**Figure 5 pharmaceutics-14-01543-f005:**
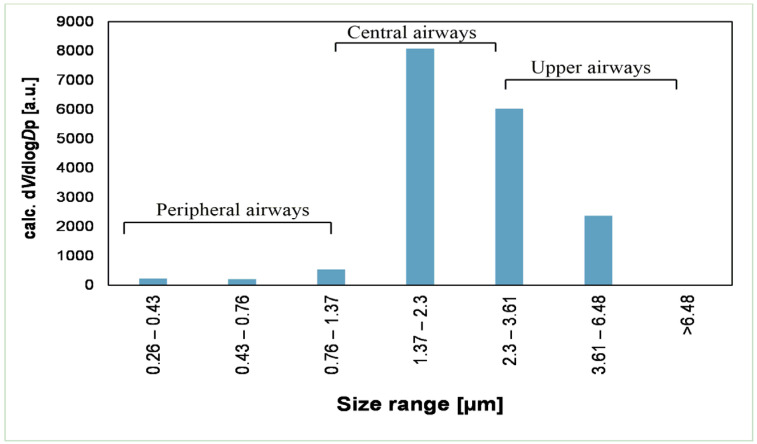
Mass size distribution of INH-MC/HA nanopowder as measured by the NGI using the data evaluation method based on the surface coverage of the collection plates.

**Figure 6 pharmaceutics-14-01543-f006:**
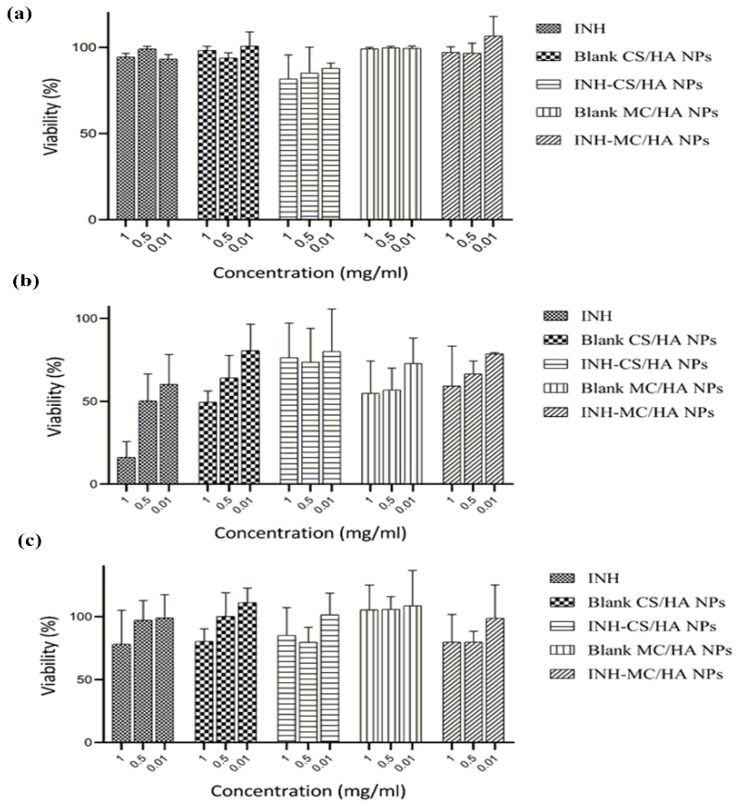
Effect of nanoformulations on the viability of A549 cells (**a**), Raw 264.7 macrophages (**b**), and primary culture of macrophages derived from human blood (**c**). The results were obtained after incubation of different concentrations of the nanoparticles (0.01, 0.5, 1 mg/mL) over 24 h. All the results were performed in triplicate and are expressed as mean ± S.D.

**Figure 7 pharmaceutics-14-01543-f007:**
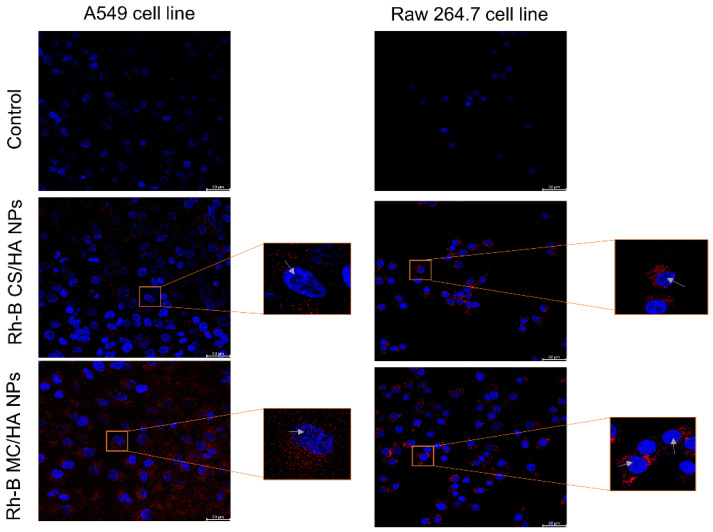
Cellular internalization study via confocal imaging. Confocal images of A549 cells and Raw 264.7 macrophages showing the internalization of Rhodamine-B labeled NPs (red) after an incubation of 2 h. The blue color indicates the nuclei staining with DAPI-dye. Rhodamine-B (Excitation λ_max_ = 546 nm, Emission λ_max_ = 568 nm), DAPI-dye (Excitation λ_max_ = 359 nm, Emission λ_max_ = 457 nm).

**Figure 8 pharmaceutics-14-01543-f008:**
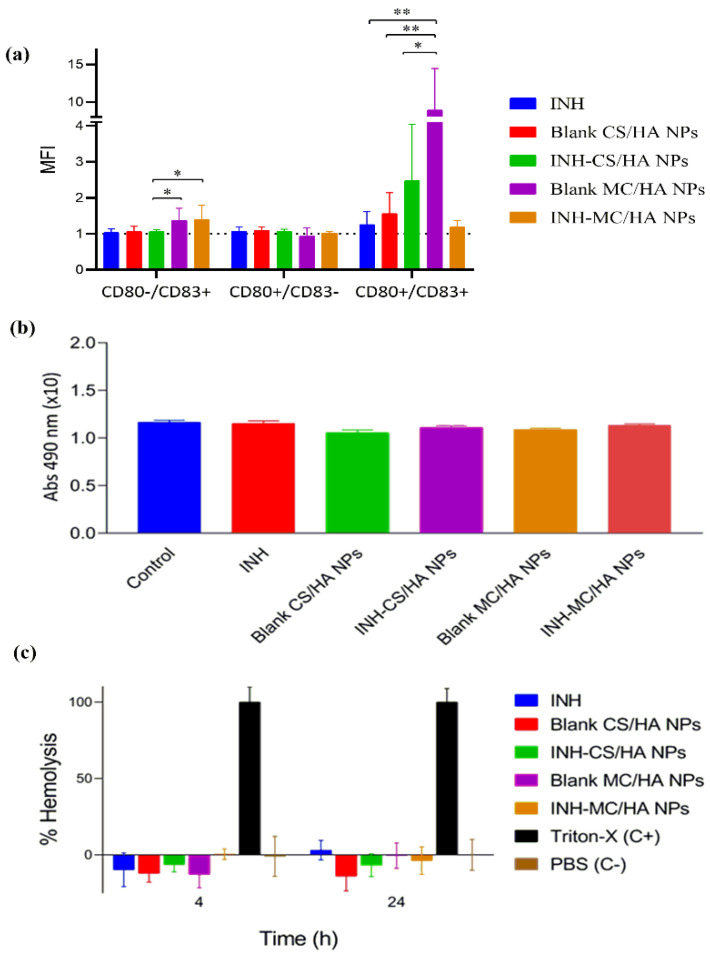
% Quantification of the expression of macrophage maturation markers, CD80 and CD83. All the results have been presented as mean fluorescence intensity (MFI) in macrophages incubated with NPs. The dotted line (MFI = 1) represents the signal from the macrophages incubated in culture (control). Data from blank MC/HA NPs were statistically significant (* *p* < 0.05, ** *p* < 0.001) (**a**), 2,3-Indoleamine dioxygenase (IDO) activity in macrophages cell culture (**b**), and in-vitro hemolysis assay after 4 and 24 h performed on fresh human blood obtained from donors after consent (**c**). All the results are expressed as mean ± SEM, *n* = 3 different blood donors (*p* < 0.001).

**Table 1 pharmaceutics-14-01543-t001:** Process parameters are taken into consideration during the process of freeze-drying.

Process	Time (h:min)	Chamber Pressure (mbar)	Product Temperature (°C)	Shelf Temperature (°C)
Freezing	01:30	-	−20	−40
	02:30		−20 to −26	
	03:45		−26 to −39	
Primary drying	04:00	0.01	−39 to −37	−25
	06:10		−37 to −31	−20
	09:40		−31 to −27	0
Secondary drying	16:00	0.01	−27 to −14	+9
	21:10		−14 to −6	+22
	40:15		−6 to −2	+30

**Table 2 pharmaceutics-14-01543-t002:** Physicochemical attributes of nanopowders.

Samples	Average Particle Size (nm)	PDI	Zeta Potential (mV)	Encapsulation Efficiency (%)	Drug Loading (%)
CS/HA NPs	310 ± 21	0.231 ± 0.12	30.3 ± 9.05	-	-
INH-CS/HA NPs	342 ± 08	0.301 ± 0.17	29.5 ± 2.01	90.18 ± 1.01	23.5 ± 1.29
MC/HA NPs	298 ± 11	0.116 ± 0.01	30.6 ± 3.79	-	-
INH-MC/HA NPs	303 ± 16	0.179 ± 0.04	34.3 ± 6.03	92.31 ± 2.06	25.9 ± 2.11

## Data Availability

Not applicable.

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
