# Peer review of "Dry Powder Comprised of Isoniazid-Loaded Nanoparticles of Hyaluronic Acid in Conjugation with Mannose-Anchored Chitosan for Macrophage-Targeted Pulmonary Administration in Tuberculosis"

_pharmaceutics, 2022, doi:10.3390/pharmaceutics14081543_

Round 1
Reviewer 1 Report
This manuscript reports the synthesis of dry powder comprised of isoniazid loaded nanoparticles of hyaluronic acid in conjugation with mannose anchored chitosan for macrophage targeted pulmonary administration in tuberculosis. It is an interesting paper especially in the preparation dry powder formulation containing nanoformulation even though there are several papers that have been published in this context. This manuscript should be considered for acceptance after minor correction.
1) Please check for typo and grammatical errors.
2) Please include drug loading data and equations.
3) Please include the relative humidity data in section 2.9 as well as in figure 3. It would be interesting to include the amount of drug that remained after the storage period.
4) How about the aerosol performance after 1 month of storage? Please include the data. How about the toxicity data after the storage? Please include the data.
5) Please add arrows to indicate the localization of internalized drug in Figure 6.
Author Response
Reviewer 1: This manuscript reports the synthesis of dry powder comprised of isoniazid loaded nanoparticles of hyaluronic acid in conjugation with mannose anchored chitosan for macrophage targeted pulmonary administration in tuberculosis. It is an interesting paper especially in the preparation dry powder formulation containing nanoformulation even though there are several papers that have been published in this context. This manuscript should be considered for acceptance after minor correction.
Response: Thank you so much for your kind comments and suggestions. We appreciate your timely response and insightful knowledge.
Question: Please check for typo and grammatical errors.
Response: Thank you. The manuscript has been checked for the mistakes.
Question: Please include drug loading data and equations.
Response: Thank you so much for the suggestion. The drug loading and equation has been added to the main text.
Question: Please include the relative humidity data in section 2.9 as well as in figure 3. It would be interesting to include the amount of drug that remained after the storage period.
Response: Thank you for your kind concern. The obtained data shown in the manuscript is about the stability of the nanosuspension before freeze-drying. Therefore, humidity was not taken in to consideration. To take into consideration the possibility of lag time in the freeze-drying process, the stability of the nanosuspensions was evaluated. However, the stability of the nanopowder obtained after freeze-drying is still going on, under different conditions of cold hot and room temperature by keeping the humidity concerns in mind. Thank you for understanding.
Question: How about the aerosol performance after 1 month of storage? Please include the data. How about the toxicity data after the storage? Please include the data.
Response: Thank you for the question. Due to the lag time in the research because of pandemic, most of the experiments were performed after the storage time of certain days. The aerosol performance did not vary to a significant level. The previously published data was obtained prior the pandemic in which MMAD and FPF <3 µm were found to be 2.98 and 50 % respectively (Mukhtar, Mahwash, et al. "Freeze-dried vs spray-dried nanoplex DPIs based on chitosan and its derivatives conjugated with hyaluronic acid for tuberculosis: In vitro aerodynamic and in silico deposition profiles." European Polymer Journal 160 (2021): 110775).
Similarly, the MTS cytotoxicity data on the human primary macrophages was performed after a lag time and the results were still found to be promising as shown in the manuscript.
On the other hand, we understand your kind concern and will try to explore the data before and after storage in the future experiments to present the obtained results with better justification.
Question: Please add arrows to indicate the localization of internalized drug in Figure 6.
Response: Thank you for the question. The localization of NPs have been highlighted using the arrows (currently figure 7).

Reviewer 2 Report
This work revealed that nano-DPI was found to be a promising vehicle for targeting macrophages. INH-MC/HA NPs were evaluated various analysis but there are unclear points in the evaluation using cells.
1. In Table 2, CS/HA NPs and INH-CS/HA NPs have larger PDI values than those of MC. Why does using MC result in monodisperse with a narrow size distribution?
2. Are Fig. 2 c and d enlarged images of parts of a and b, respectively, or are they completely different locations? Please add explanations in the image and in the legend.
3. Are Fig. S5 and Fig. 4 showing particle size distribution a single measurement? Please be sure to accurately state the number of measurements (n=X) and error bars.
Also, for Table S1, which is the result of four measurements, state the S.D. or S.E. In addition, please arrange the appearance as a table.
4. Is the 436th line of text in the process of being written?
5. Why does Blank appear to be more toxic in CS / HA NPs than in the INH-encapsulated group in terms of the survival rate of Raw 264.7 macrophages?
6. The primary culture of macrophages derived from human blood was used in this study. Is the expression of mannose receptor in Raw 264.7 macrophages and primary cultured cells similar?
7. The authors described that the internalization of NPs in the A549 cells was lower as compared to the Raw 264.7 macrophages (lines 449-450). You can enlarge a part of the image to make it easier to understand.
8. The authors also described that the internalization intensity of the NPs was found to be higher in the case of MC/HA NPs as compared to CS/HA NPs demonstrating the advantage of mannose conjugation to the polymer (lines 455-457). Please quantify this objectively with multiple images to show this. In particular, the flow cytometry results in the supplement do not successfully show the significance of MC/HA NPs over CS/HA NPs, although the authors also describe filtering issues.
9. In cellular internalization study in A549 cells, what appears to be fluorescent in areas other than the cytoplasm (where there are no cells)?
10. In the results description of Fig. 7a of lines 472-473, the expression of CD83 or CD80 was many folds higher for the MC/HA NPs when compared to other nanoformulations. Fig. 7a should show the results of the statistical analysis in the graph. Also, in the lines 476-477, blank and drug-loaded MC/HA NPs significantly upregulated the co-stimulatory markers in comparison to other NPs. Please show the results of the statistical analysis for this as well in the graph. In any case, please clearly state the control group.
11. In Fig. 7a, why do the INH-MC/HA NPs in the CD80+/CD83+ group appear to have reduced MFI compared to blank?
12. In hemolytic activity of Fig. 7c, many groups show negative hemolytic activity. Do NPs have a protective effect on the membrane?
13. Please enter the approval number of the ethics committee in the manuscript.
Throughout the manuscript
14. Please unify the use of n=X and triplicate as they are mixed up. Is there a reason why they are used differently?
15. Please revise the appearance of the graph, including supplements. cf. Remove lines in graph area, add axis and scale lines.
Author Response
Reviewer 2: This work revealed that nano-DPI was found to be a promising vehicle for targeting macrophages. INH-MC/HA NPs were evaluated various analysis but there are unclear points in the evaluation using cells.
Response: Thank you so much for your kind comments and suggestions. We appreciate your timely response and insightful knowledge
- In Table 2, CS/HA NPs and INH-CS/HA NPs have larger PDI values than those of MC. Why does using MC result in monodisperse with a narrow size distribution?
Response: Thank you for the comment. There was a slight difference in the zeta potential values of the chitosan and mannosylated chitosan based NPs. Due to high zeta potential values, the NPs were monodisperse in nature. Correlating, PDI values for MC based NPs were comparatively lower as the high particle surface charge imparts stability to the formulations.
- Are Fig. 2 c and d enlarged images of parts of a and b, respectively, or are they completely different locations? Please add explanations in the image and in the legend.
Response: Thank you for the comment. The figures 2 c,d and a,b are the different images with different scale bars, taken separately from SEM as indicated in the micrographs. The caption has been modified.
- Are Fig. S5 and Fig. 4 showing particle size distribution a single measurement? Please be sure to accurately state the number of measurements (n=X) and error bars. Also, for Table S1, which is the result of four measurements, state the S.D. or S.E. In addition, please arrange the appearance as a table.
Response: Thank you for the comment. Figure S5 presents outcome from four measurements to determine MMAD and FPF accurately prior to using NGI. Error bars have been incorporated. Figure 5 (previously 4) demonstrates the size distribution in a single measurement. Table S1 has been arranged in a table format with S.D.
- Is the 436th line of text in the process of being written?
Response: Thank you for the response. The sentence is complete, the punctuation mark, full stop, was missing at the end of the sentence.
- Why does Blank appear to be more toxic in CS / HA NPs than in the INH-encapsulated group in terms of the survival rate of Raw 264.7 macrophages?
Response: We highly appreciate this observation. The MTT assay allows to evaluate the cellular metabolic activity. This oxidative metabolism is greater when a situation of cellular stress occurs. The influence of cationic nanocapsules on the metabolic activity of macrophages has been reported, as they have better electrostatic properties, leading to a greater attraction with the negatively charged cell surface (S. Robla, M. Prasanna, R. Varela-Calviño, C. Grandjean, N. Csaba, A chitosan-based nanosystem as pneumococcal vaccine delivery platform, Drug Deliv. Transl. Res. 11 (2021) 581–597. https://doi.org/10.1007/s13346-021-00928-3). This property, together with the dose-dependent character of the formulations would explain a more pronounced decrease in cell viability for Blank NPs compared to those loaded in RAW 264.7 cells, since they are the ones with a higher positive charge.
- The primary culture of macrophages derived from human blood was used in this study. Is the expression of mannose receptor in Raw 264.7 macrophages and primary cultured cells similar?
Response: We acknowledge reviewer for this observation. The mannose receptor (MR) is expressed in abundance on alveolar macrophages, monocyte-derived macrophages (MDMs), and dendritic cells. However, several macrophages’ lines (as HL60, U937, THP-1) do not express the exact surface markers that are seen on primary macrophages. As MR-positive cell lines are of murine or rat derivation (D.J. Vigerust, S. Vick, V.L. Shepherd, Characterization of functional mannose receptor in a continuous hybridoma cell line, BMC Immunol. 13 (2012) 1. https://doi.org/10.1186/1471-2172-13-51), we selected the murine macrophage Raw 264.7 cell line which have been extensively used in studies of MR function for our experiments.
- The authors described that the internalization of NPs in the A549 cells was lower as compared to the Raw 264.7 macrophages (lines 449-450). You can enlarge a part of the image to make it easier to understand.
Response: We thank the referee for these observations. Figure 6 was optimized for better understanding. Cells without Rhodamine B NPs were used as a control to adjust the detector gain and establish the baseline.
- The authors also described that the internalization intensity of the NPs was found to be higher in the case of MC/HA NPs as compared to CS/HA NPs demonstrating the advantage of mannose conjugation to the polymer (lines 455-457). Please quantify this objectively with multiple images to show this. In particular, the flow cytometry results in the supplement do not successfully show the significance of MC/HA NPs over CS/HA NPs, although the authors also describe filtering issues.
Response: We would like to thank the reviewer for the comment. Figure 6 has been further zoomed for better understanding.
- In cellular internalization study in A549 cells, what appears to be fluorescent in areas other than the cytoplasm (where there are no cells)?
Response: Supplementary figure S5 shows merged bright-field (first row) and fluorescent confocal images (second row) acquired for CS and MC nanoparticles after incubation with the A549 cell line. The blue channel corresponds to DAPI-stained nuclei, while the red fluorescence channel corresponds to NPs labelled with rhodamine B, respectively. As shown in the second row, the nanoparticles are located surrounding cell nuclei, however they are also found within the cytoplasm of alveolar macrophages, as confirmed in the left column. Bright-field analysis allows us to delimit the different cells, confirming that an internalization or adsorption of the NPs takes place in the plasma membrane. Control image taken in transmission as observed using white-light illumination.
- In the results description of Fig. 7a of lines 472-473, the expression of CD83 or CD80 was many folds higher for the MC/HA NPs when compared to other nanoformulations. Fig. 7a should show the results of the statistical analysis in the graph. Also, in the lines 476-477, blank and drug-loaded MC/HA NPs significantly upregulated the co-stimulatory markers in comparison to other NPs. Please show the results of the statistical analysis for this as well in the graph. In any case, please clearly state the control group.
Response: Thank you for the comment. The statistical analysis have been mentioned in the graph. Control has also been mentioned.
- In Fig. 7a, why do the INH-MC/HA NPs in the CD80+/CD83+ group appear to have reduced MFI compared to blank?
Response: Thank you for your question. It is true that the exposure of the INH-MC NPs to immature monocyte-derived macrophages did not alter the cell phenotype, as shown in the co-stimulatory expression levels of the cell surface molecules CD80 and CD83. However, it is not indicative that the NPs have not been recognized by the macrophages, because as confirmed by confocal images, this exposure resulted in NPs uptake by the cells. As shown in the first group, we can observe that MC increased the expression of CD83+ (CD80-/CD83+) surface molecules. However, in the case of INH, no differences were observed in the expression of CD80 (CD80+/CD83-). Similar observations have been described for isoniazid mannosilated NPs (J. Pi, L. Shen, E. Yang, H. Shen, D. Huang, R. Wang, C. Hu, H. Jin, H. Cai, J. Cai, G. Zeng, Z.W. Chen, Macrophage-Targeted Isoniazid–Selenium Nanoparticles Promote Antimicrobial Immunity and Synergize Bactericidal Destruction of Tuberculosis Bacilli, Angew. Chemie - Int. Ed. 59 (2020) 3226–3234. https://doi.org/10.1002/anie.201912122) (N. Khan, L. Mendonca, A. Dhariwal, G. Fontes, D. Menzies, J. Xia, M. Divangahi, I.L. King, Intestinal dysbiosis compromises alveolar macrophage immunity to Mycobacterium tuberculosis, Mucosal Immunol. 12 (2019) 772–783. https://doi.org/10.1038/s41385-019-0147-3). Besides, GM-CSF induces M1 type macrophage polarization which express low levels of CD80 marker compared with M2 polarized macrophages (J.C. Zarif, J.R. Hernandez, J.E. Verdone, S.P. Campbell, C.G. Drake, K.J. Pienta, A phased strategy to differentiate human CD14+ monocytes into classically and alternatively activated macrophages and dendritic cells, Biotechniques. 61 (2016) 33–41. https://doi.org/10.2144/000114435).
- In hemolytic activity of Fig. 7c, many groups show negative hemolytic activity. Do NPs have a protective effect on the membrane?
Response: Thank you for the comment. The NPs can be hemolytic in the dose dependent manner. Correlating with the cytotoxicity assay, the NPs concentration chosen for the hemolytic activity was chosen to be 0.01 mg/ml. Moreover, the NPs were incubated for the maximum duration of 24 h. However, if the incubation time increases, the hemolysis also increases. Furthermore, no such data exists which prove that NPs are hemo-protective. They were simply not hemolytic in lower concentration and shorter incubation time.
- Please enter the approval number of the ethics committee in the manuscript.
Response: Approval number 2014/543
- Please unify the use of n=X and triplicate as they are mixed up. Is there a reason why they are used differently?
Response: Thank you for the concern. All the experiments were performed in triplicate unless otherwise stated. The term triplicate was used for the number of experimental runs. However, in case of macrophage phenotype study, blood was obtained from 3 different donors, thereby referring as n=3 different blood donors.
- Please revise the appearance of the graph, including supplements. cf. Remove lines in graph area, add axis and scale lines.
Response: Thank you. Changes have been made.

Reviewer 3 Report
Dear Authors,
I have to admit that the presented work is really interesting and provides not just scientific but also practical value. Tuberculosis has recently become a bigger problem around the world, and effective therapy is challenging both for medical practitioners and the pharmaceutical industry in terms of getting effective dosage forms.
It is worth mentioning that the Authors provided not just technological process but also assessed safety by cytotoxicity studies and hemolysis effect as well as the effect on macrophages.
I suggest to build-up an introduction part related to other reported formulations for potential use in TB therapy using pulmonary delivery systems. There are just a few sentences, but in light of the presented results, it will be good to extend them.
I suggest implementing figure S2 to the main text. The applied method of aerodynamic properties assessment with a breath simulator is still innovative and definitely worth bo be presented in chapter 2.10
The freeze-drying process to obtain the DPI is getting attention. One issue I always think about is the possibility of further operations without affecting particle structure. If I understand correctly after lyophilization was obtained product was crushed/milled? Do you see any difficulties with using freeze-dried products for further processing for example by adding a classical carrier to them?
Kind regards
Author Response
Reviewer 3: I have to admit that the presented work is really interesting and provides not just scientific but also practical value. Tuberculosis has recently become a bigger problem around the world, and effective therapy is challenging both for medical practitioners and the pharmaceutical industry in terms of getting effective dosage forms. It is worth mentioning that the Authors provided not just technological process but also assessed safety by cytotoxicity studies and hemolysis effect as well as the effect on macrophages.
Thank you so much for your kind comments and suggestions. We appreciate your timely response and scientific contribution to making this manuscript refined.
Question: I suggest to build-up an introduction part related to other reported formulations for potential use in TB therapy using pulmonary delivery systems. There are just a few sentences, but in light of the presented results, it will be good to extend them.
Response: Thank you. Few studies have been included and highlighted in the introduction related to TB therapy using pulmonary delivery systems.
Question: I suggest implementing figure S2 to the main text. The applied method of aerodynamic properties assessment with a breath simulator is still innovative and worth to be presented in chapter 2.10
Response: Thank you. Figure S2 has been placed in the main text according to your suggestion. The applied method has also been mentioned in section 2.10.
Question: The freeze-drying process to obtain the DPI is getting attention. One issue I always think about is the possibility of further operations without affecting particle structure. If I understand correctly after lyophilization was obtained product was crushed/milled? Do you see any difficulties with using freeze-dried products for further processing for example by adding a classical carrier to them?
Response: Thank you again. The obtained nanosuspension (via ionic gelation method) was freeze-dried using trehalose as a cryoprotectant. So, the powder obtained after freeze-drying was not crushed or milled (as it was a bottom-up technique). The obtained nanopowder was a fine powder and the process of freeze-drying is industrially being used to obtain the dry powders for pulmonary or parenteral administration. Trehalose helps in improving the powder properties by reducing the rate of particle aggregation during drying and storage. We did not face any difficulty while performing any studies with the lyophilized powders.

Round 2
Reviewer 2 Report
The authors have substantially revised the manuscript to address reviewer comments.